# Subcellular mass spectrometry imaging of lipids and nucleotides using transmission geometry ambient laser desorption and plasma ionisation

Reuben S. E. Young [1], Ann-Katrin Piper[1], Luke McAlary[1], Jayden C. McKinnon[1], Jeremy S. Lum [2], Jens Soltwisch [3], Marcel Niehaus [4] & Shane R. Ellis [1] ✉

Demand for mass spectrometry imaging (MSI) technologies offering sub-cellular resolution for tissues and cell imaging is rapidly increasing. To accomplish this, efficient analyte ionisation is essential, given the small amounts of sample material in each pixel. Herein, we describe an atmospheric pressure transmission-geometry matrix-assisted laser desorption source equipped with plasma ionisation. By utilising a pre-staining method for sample preparation, lipid signal intensities were enhanced by an order of magnitude compared to conventional matrix-only methods, while serendipitously enabling imaging of numerous nucleotides. The system enables detection of up to 200 lipids and nucleotides in tissues at 1 μm-pixel size while informative MSI data is still obtained down to 250 nm pixel size. The use of sub-micron pixels is shown to discern subcellular features through combinations with fluorescence microscopy. This method provides a powerful tool for cellular and sub-cellular imaging of small molecules from tissues and cells for spatial biology applications.

Eukaryotic tissues comprise numerous cell types—each discretely organised and kitted with unique physical and chemical properties to fulfil complex functions. One class of biomolecule that contributes to these cellular physicochemical differences, including to membrane structural properties and biochemical signalling events, is lipids. The multifunctional nature of these molecules is owed to a vast diversity in their molecular structures, and subsequently cellular lipid profiles are known to differ between cell types[1,2] and disease states (e.g. cancers[3–5], osteoarthritis[6], motor neurons disease[7] and Parkinson's disease[8]). While there are numerous analytical techniques capable of discerning these molecular level differences in homogenised tissue and cell culture extracts[9–12], these methods are generally blind to heterogeneity within cell populations and fail to assign specific lipid profiles to

discrete cells or tissue regions. Thus, methods capable of retaining this level of information are highly desirable for spatial biology and medical research fields.

Mass spectrometry imaging (MSI) enables label-free imaging of many analyte classes within tissues and cells[13–19]. The most popular method is matrix assisted laser desorption/ionisation (MALDI), which uses a UV laser to sample material across a matrix coated sample. This photoactive matrix promotes the concerted laser desorption of analytes and their subsequent ionisation[20]. This ionisation process, however, is inefficient and typically results in detection biases towards the most ionisable molecular species, with ionisation efficiencies that can differ up to 3–4 orders of magnitude[21–23]. Recent advances saw the advent of laser-induced post ionisation (MALDI-2)[24], whereby a

[1]Molecular Horizons and School of Science, University of Wollongong, Wollongong, NSW, Australia. [2]Molecular Horizons and School of Medical, Indigenous and Health Sciences, University of Wollongong, Wollongong, NSW, Australia. [3]Institute of Hygiene, University of Münster, Münster, Germany. [4]Bruker Daltonics GmbH & Co. KG, Bremen, Germany. ✉e-mail: sellis@uow.edu.au

secondary laser pulse is focussed into the plume generated by the initial MALDI laser pulse. This generates an additional MALDI-like event that improves ion yields by up to two orders of magnitude[15,24]. Given the inverse relationship between spatial resolution and sampling volume, this sensitivity gain has further advanced MALDI-MSI spatial resolution.

This advance has greatly benefited transmission geometry MALDI (t-MALDI), which can be used for high spatial resolution molecular imaging at pixel sizes as low as 1 μm. First described in the context of MALDI-MSI by Zavalin et al.[25], t-MALDI focusses the laser through the rear of a glass slide, allowing use of higher numerical aperture optics without encumbering the space needed for ion optics. While ablation crater sizes of 1–2 μm were achieved, detection of biomolecules such as lipids was limited to only several abundant species. In 2019, Niehaus et al. demonstrated that by combining MALDI-2 with t-MALDI, lipid coverage was significantly enhanced and more than 30 unique lipid sum composition species were visualised in mouse brain at 1 μm pixel sizes[26]. t-MALDI-2 has also been applied to single cell MSI[27]. Alternative post-ionisation strategies for improved detection have also been explored. In 2020, Elia et al. introduced an inline dielectric barrier discharge device to an atmospheric pressure (AP) t-MALDI source and demonstrated 5 μm MSI capabilities[28]. Similarly, Bookmeyer et al. demonstrated up to a 100-fold increase in signal intensity by combining krypton lamps with MALDI-MSI[29]. A similar single-photon approach used by Qi et al. combined krypton lamps with AP transmission geometry laser desorption to achieve 4 μm pixel sizes[30]. Although these advances have been substantial, combining micron and sub-micron spatial resolutions with broad analyte detection has remained elusive, thus limiting MSI studies of subcellular structures.

Herein, we report a custom-built transmission-geometry, AP-MALDI source with plasma ionisation that is capable of <1 μm MSI pixel size (i.e. t-MALDI-P), and a non-conventional sample preparation method that not only significantly enhances signal intensities and coverage for lipids but serendipitously enables MSI of nucleotides. This presents a significant advance in capability for ultra-high spatial resolution MSI. A general overview of this source build is displayed in the Fig. 1 schematic (Fig. 1a) and photographs (Fig. 1b–d). Briefly, a frequency-tripled 355 nm laser pulse (Fig. 1a vi) is shaped by an 8× Keplerian geometry telescope (Fig. 1a vii) before being reflected by a dielectric mirror (Fig. 1a ii) through a 50× infinity-corrected microscope objective (Fig. 1a iii) and transmitted through the rear of a glass microscopy slide. Concurrently, a camera (Fig. 1a i) captures the reflected visible light from the sample slide and serves as a sample visualisation optic (cf. Fig. 1b). The sample slide is mounted on a three-axis piezoelectric stage (Fig. 1a iv) with nanometre positioning. Laser ablation events lead to the ejection of analyte species, which are drawn through a heated steel capillary (~350 °C) using the vacuum of a modified timsTOF Pro mass spectrometer, through the SICRIT plasma device (Fig. 1a v). Neutral analytes are soft-ionised indirectly by reactive species generated by the cold-plasma device[31], before mass spectrometric analysis. For further experimental details and parameters see 'Methods': AP-t-MALDI-P.

## Results

### Adapted sample preparation for improved analyte detection in tissues

MALDI-MSI requires the homogenous deposition of matrix onto the sample. For high resolution applications, where crystal sizes need to be minimised, this is commonly achieved using sublimation-based matrix deposition. Figure 2A displays a conventional workflow used for MALDI-MSI, including matrix sublimation (Fig. 2a i), MSI (Fig. 2a ii), histological staining (Fig. 2a iii) and microscopy (Fig. 2a iv–v). To demonstrate the capabilities of t-MALDI-P using this conventional sample preparation with 4-(dimethylamino)cinnamic acid (DMACA) as

the matrix, we applied this method to sagittal sections of murine brain tissue and imaged the hippocampal regions at 2 μm pixel size (Fig. 2b) and 10× brightfield microscopy (Fig. 2c). Concurrently, to investigate ablation crater size for the conventional preparation approach, 5 μm-spaced ablation arrays were obtained using MSI relevant laser energies (cf. 'Methods' section) for scanning electron microscopy (SEM), which revealed ~1.75 μm wide ablation craters (Fig. 2a v).

While MSI and microscopy can be conducted on the same sample, due to the semi-destructive nature of t-MALDI experiments, staining and microscopy for Fig. 2a–c was undertaken on an adjacent serial section to preserve tissue section quality for microscopy. However, at high spatial resolutions, image co-registration of serial sections can be poor due to section thickness being approximately the width of a cell and thus images display different cell layers. To circumvent this, we adapted conventional preparation methods. The tissue is first stained with cresyl violet (CV, Fig. 2d i) for microscopy (Fig. 2d ii), before matrix sublimation (Fig. 2d iii), MSI (Fig. 2d iv) and SEM (Fig. 2d v). To preserve lipids, we adapted the CV protocols to avoid de-lipidation/permeabilisation by instead submerging the sample under a drop of 1% CV acetate solution and removed excess with aqueous ammonium fluoride. This adapted method, herein referred to as 'pre-staining', was extensively tested for displacement or translocation of the sample material(s) and delocalisation of lipids, of which none could be observed (cf. Supplementary Fig. 1). This method was applied to an equivalent hippocampus section and imaged using 2 μm MSI (Fig. 2e) and 10× brightfield microscopy (Fig. 2f).

Surprisingly, when using identical laser energies as the conventional preparation to generate 5 μm-spaced ablation arrays for the pre-staining approach, SEM revealed notably smaller ablation crater sizes from the pre-stained sample (Fig. 2d v; ~750 nm). Thus, the laser energy was adjusted to maximise the ablated material from a 2 μm pixel (i.e. ablation crater size ≤ 2 μm). Increasing laser energy, while maintaining comparable ablation crater diameters, yielded a pronounced enhancement in analyte ion yield—up to an order of magnitude—evident in the increased mass spectral signal intensity observed in the pre-stained tissue sample (Fig. 2h) relative to the conventionally prepared sample (Fig. 2g). To exclude that this effect was related to the ammonium fluoride salt wash, Supplementary Fig. 2 demonstrates that the same signal intensity increase can be observed even when using a water wash for CV stained tissues. Post-MSI microscopy suggests that this boost to signal intensity is due to increased sample ejection, with micrographs displaying more extensive tissue loss for the pre-stained tissue experiment (Supplementary Fig. 3). To observe the effect the laser had beneath the matrix surface, focused ion beam (FIB)-SEM was conducted on the ablation arrays (cf. Supplementary Fig. 4). This allowed for cross-sectional visualisation of the ablation craters and revealed that pre-staining brought about cavitations and the absence of a 'column' of tissue material within the path of the laser. Conversely, only the surface (i.e. matrix layer) was ablated without the presence of CV within the tissue. At 2 μm pixel sizes, this ablation column represents ~8 pg of material, thus maximising the effective sampling efficiency of such low analyte volumes is highly desirable for high spatial resolution MSI experiments.

The effect of improved sampling efficiency becomes apparent when contrasting the averaged mass spectra, which reveal up to a 10-fold increase in ion intensity from the pre-stained tissue (Fig. 2h) compared to the unstained tissue (Fig. 2g)—an effect that was reproducible for alternative matrix choices (cf. Supplementary Fig. 5). Similarly, the spatial distributions and intensities of 5 lipids across both MS images (Fig. 2b, e; Supplementary Fig. 6) reveal that pre-staining greatly improves ion intensity and detection efficiency (e.g. the observation of Cer 42:2;2 + H+ [magenta] in Fig. 2e, but absence from Fig. 2b). Additionally, the combination of plasma ionisation and increased sensitivity translates to the observation of protonated anionic lipid classes, such as phosphatidylserine (PS) and

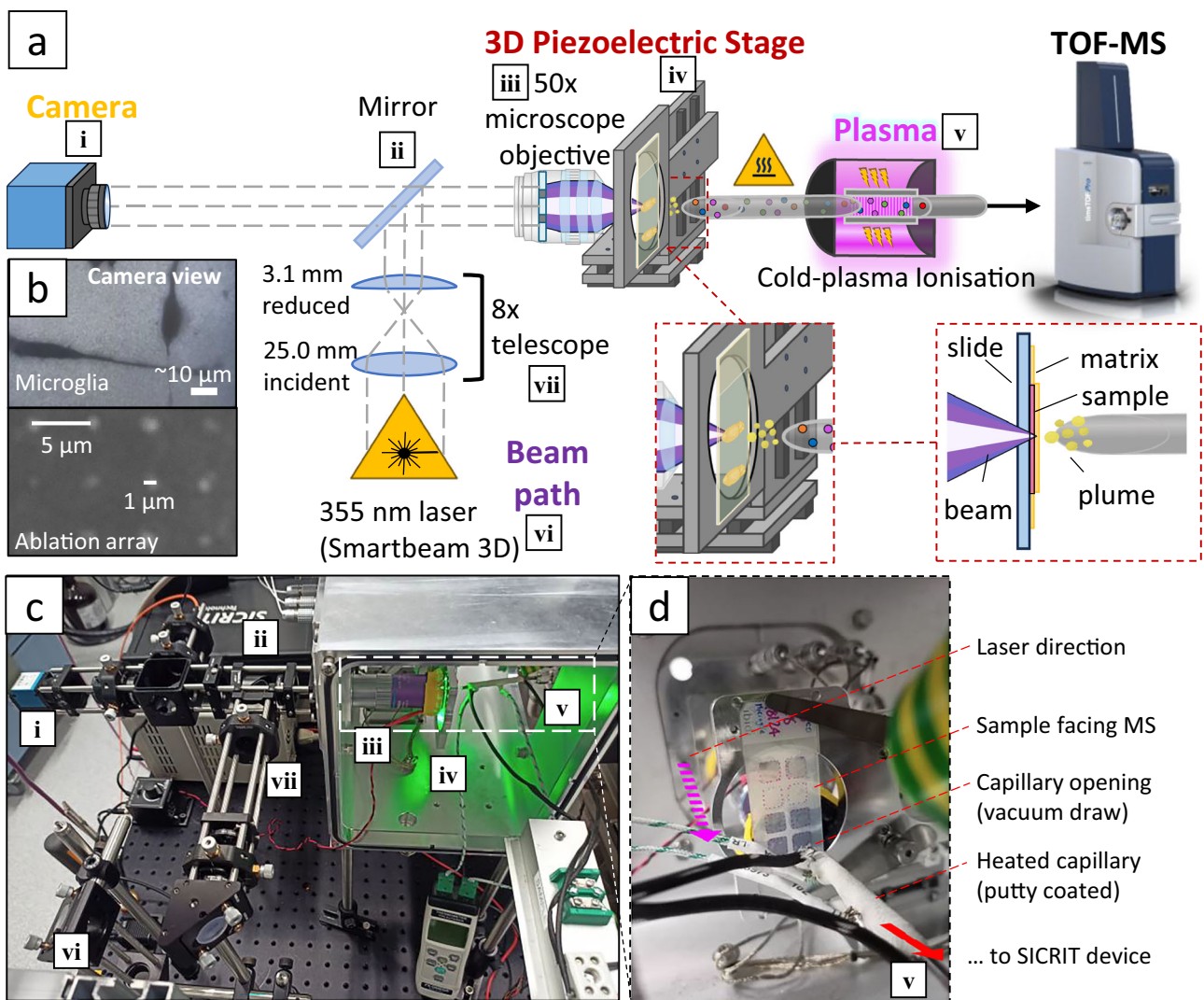

**Fig. 1 | t-MALDI-P source schematic and build. a** Schematic showing the optical and analyte paths of the t-MALDI-P setup, including optical microscopy (i, ii, iii), laser path including beam shaping (vi, vii, ii, iii), analyte desorption (iv) and ion generation using cold-plasmas (v). Ions generated by the SICRIT plasma device are then drawn into the mass spectrometer for detection. **b** Images showing a microglia cell sample, and a laser ablation array as viewed through the onboard microscopy camera optic. Precise piezoelectric stage movements allow for spot size to be roughly estimated from the lateral movement distance. **c** Physical build of the source depicting the components listed in (**a**). **d** Close-up view of the sample stage and components, showing the direction of the laser (purple arrow) and the direction of the ions through the heated capillary (red arrow).

phosphatidylinositol (PI) in Fig. 2d, which are rarely observed as protonated species in conventional MSI. Using a semi-automated lipid identification pipeline, the lipids identified and semi-quantified from the hippocampus MSI data are displayed in the pie chart inserts in Fig. 2g, h and reveal a total of 164 lipid sum composition species from the unstained and 213 from the pre-stained tissues meeting inclusion criteria for ID (cf. Methods and Supplementary Tables 1, 2). As has previously been identified with plasma ionisation[32], phosphatidylcholine (PC) and ether-PC (PC-O) undergo in-source fragmentation and characteristic loss of the phosphocholine headgroup while still retaining fatty acyl sum composition information (note that other phospholipids can also undergo similar fragmentation, but this has been shown to be dominated by PC)[32]. To indicate lipid identifications that were made using these fragment ions, throughout, we have used inverted commas around the lipid headgroup nomenclature to represent that they predominantly come from PC (e.g. 'PC' 34:1).

Interestingly, a non-lipid ion at $m/z$ 342.123 was found localised in the granule cells of the dentate gyrus in the pre-stained tissue (Fig. 2e; cyan). Tandem mass spectrometry (MS²) revealed the identity of this ion

to be CV bound to a dehydrated ribose sugar. Synthetically derived ribonucleotide standards of adenine (A), guanine (G), cytosine (C) and uracil (U) were combined with CV for MALDI experimentation and revealed that $m/z$ 342 was specific formed for purine ribonucleotides (A/G) but not pyrimidines (C/U). Interestingly, a colour change from purple to blue was observed when mixing CV with the purines, suggesting that conjugation had been altered. While the binding mechanism of CV to Nissl bodies (i.e. neuronal subcellular regions enriched in rough endoplasmic reticulum and ribosomes) is largely unclear[33], we hypothesise that CV is covalently binding to purine ribonucleotides at the ribosomes, which subsequently forms cations during plasma ionisation. Additionally, intact monophosphoribonucleotides and monophosphodeoxyribonucleotides were present in the MSI as dehydrated cations (i.e. [M + H-H₂O]⁺), with their nuclear or ribosomal cellular origins being distinguishable by the presence or absence of a carbon-carbon double bond, likely formed through the elimination of water from the sugar moiety. For further information about nucleotide structures, synthetic standards MS and tandem MS cf. Supplementary Figs. 7, 8).

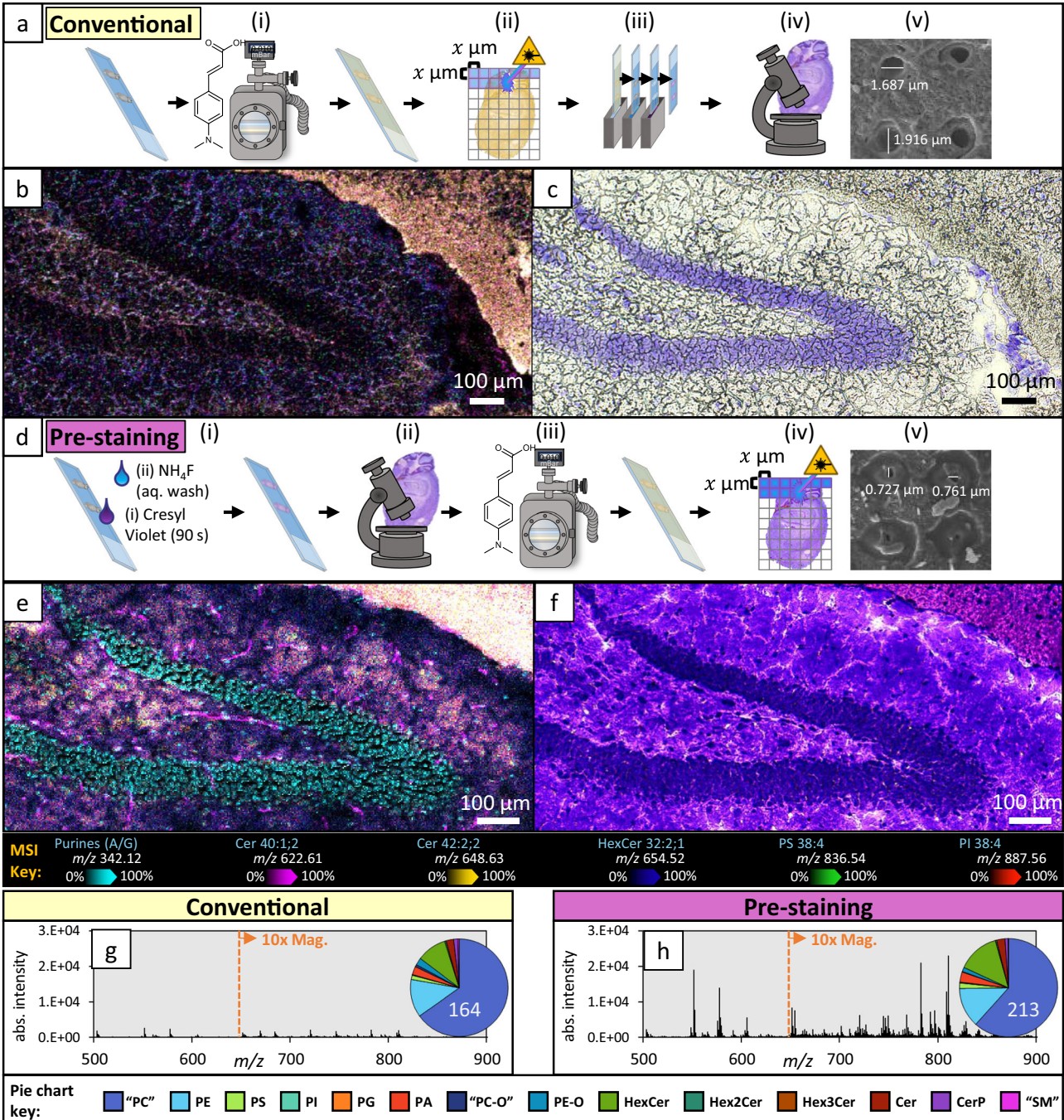

**Fig. 2 | Adapted sample preparation method developed for t-MALDI-P and applied to sagittal murine hippocampus sections. a** Conventional MALDI preparations involve the deposition of a photoactive matrix (i), MSI (ii), histological staining (of same or serial sections; iii) and optical microscopy (iv). Post-ablation scanning electron microscopy can then be taken to confirm ablation crater sizes (v). Two serial sagittal-sections of mouse hippocampus were imaged using the conventional method (**a**), with one section being imaged using 2 μm pixel size (**b**) and the other undergoing conventional CV staining methods before 10× brightfield microscopy was obtained (**c**). **d** The newly developed method first stains the sample (i) to allow for high-quality microscopy (ii) to be obtained prior to the deposition of MALDI matrix (iii) and MSI experiments (iv). Post-ablation scanning electron microscopy can then be taken to confirm ablation crater sizes (v). **e**, **f** Using the pre-staining method **d** allows for a single sagittal-section of mouse hippocampus to be imaged through both modalities. **e** Displays MSI at 2 μm pixel size and **f** displays 10× brightfield microscopy of the same section, which had been pre-stained with CV. **g**, **h** The averaged mass spectra from the total imaging areas in (**b**) and (**e**), corresponding to (**h**) and (**i**), respectively. For ease of comparison, the same y-axis scale is used, and a 10× magnification has been applied to peaks $m/z$ 650 < x < 900. Pie chart inserts show the total lipid species identified from either spectrum with spectral intensities being grouped by class. Non-overlaid MS ion images for either experiment can be found in Supplementary Fig. 6. All microscopy images were obtained in triplicate ($n = 3$) and display similar results.

## Analyte coverage at 1 μm pixel size

Matrix selection is known to impact desorption and ionisation within MALDI experiments[34]. To determine the ideal matrix for high spatial resolution MSI using t-MALDI-P, we conducted 1 μm pixel size MSI on the secondary fissure of murine cerebellum for three matrices using the pre-staining technique: 2,5-dihydroxyacetophenone (DHA), α-cyano-hydroxycinnamic acid (CHCA) and 4-(dimethylamino)cinnamic acid[35] (DMACA). It was concluded that DMACA allowed for ~3-

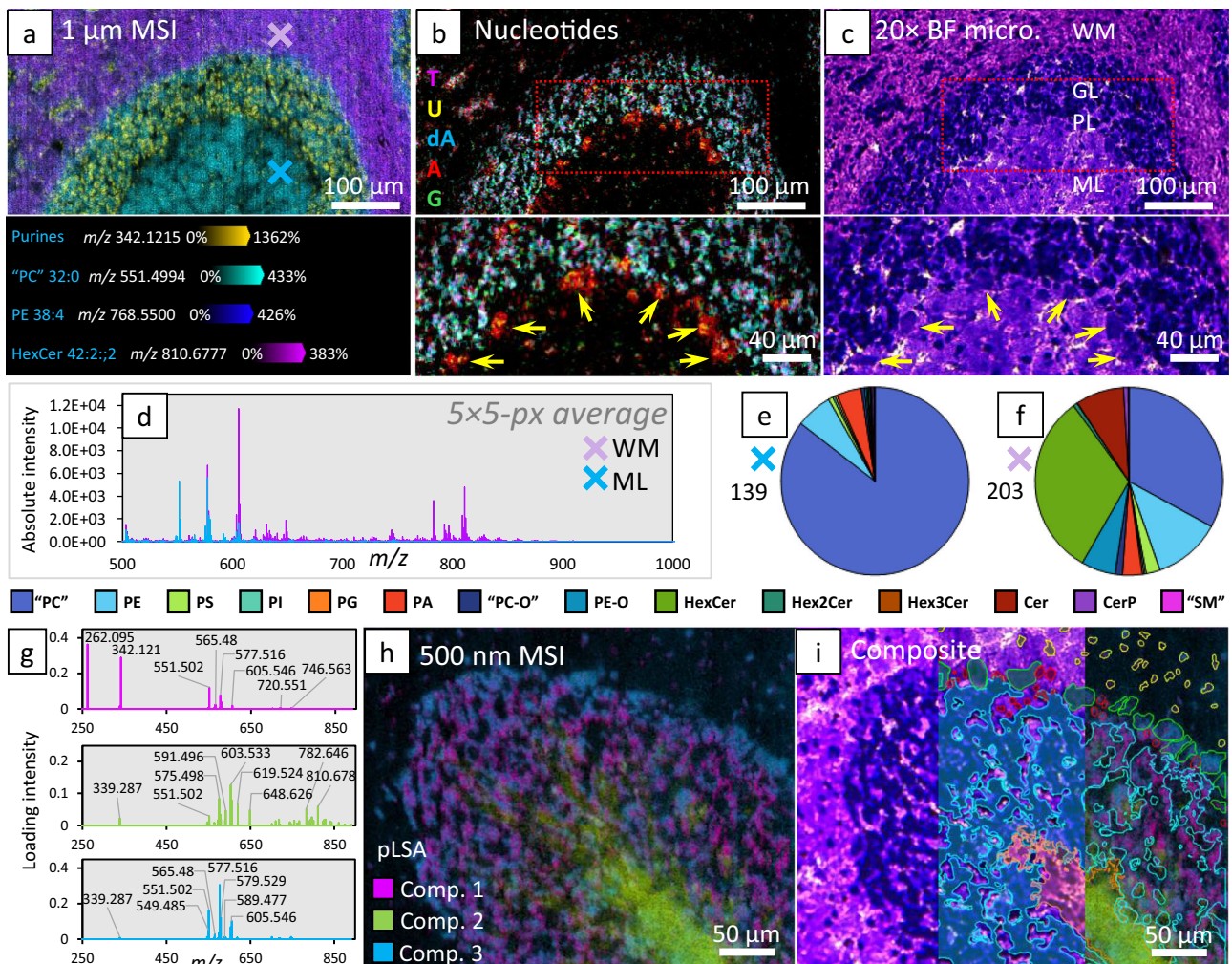

**Fig. 3 | Analyte coverage within sagittal murine cerebellar cortex sections at micron and sub-micron pixel sizes. a** 1 μm MSI of the secondary fissure between the Pyramus (VIII) and Uvula (IX) lobules in the cerebellar vermis. Figure panel displays an overlay of 4 ion images, including a CV complex with purine nucleotides (adenine and guanine [A/G]), 'PC' 32:0, PE 38:4 and HexCer 42:2;2. DMACA sublimation was used. **b** Figure panel displays an overlay of five deoxyribo- or ribonucleotides, namely: thymidylate (T; purple) *m/z* 305.05, uridylate (U; yellow) *m/z* 307.03, deoxyadenylate (dA; blue) *m/z* 314.06, adenylate (A; red) *m/z* 330.06 and guanine adducted to CV (G; green) *m/z* 412.13. **c** The pre-stained tissue imaged through 20× brightfield microscopy. Magnifications of the red boxes in B and C highlight the localisation of adenine and uracil to the Purkinje cells (yellow arrows). **d** Averaged (5 × 5 pixels) mass spectra from WM and the ML (indicated by purple and blue 'X') from the MS image. Graphical comparison of total lipid species identified in the blue 'X' (ML; **e**) and the purple 'X' (WM; **f**) regions. Lipid classes

monitored in the pie-charts are annotated in the key and lipid species counts are indicated beside each chart. The loading spectra of 3 components of a 5 component pLSA of 500 nm MSI from the Declive (VI) lobule of murine cerebellar cortex (**g**) and the corresponding pLSA MSI at 500 nm pixel size (**h**). **i** Using the 20× microscopy obtained prior to MSI, cell types and tissue regions can be identified and grouped using QuPath tools and subsequently mapped back on to the MSI image in SCiLS Lab for independent analysis. Enclosed regions approximately correspond to glial cells in the outer molecular layer (yellow), pyriform Purkinje neurons (green), Bergmann protoplasmic astrocytes (red), granule neurons (blue) and cerebellar white matter (orange). Differences in the averaged mass spectra from each region can be found in the supporting information (Supplementary Fig. 11). All microscopy images were obtained in triplicate (*n* = 3). (WM white matter, GL granular layer, PL Purkinje layer, ML molecular layer).

fold higher signal intensity than DHA and up to 10-fold higher signal than CHCA (cf. Supplementary Fig. 9).

Examples of lipid ion images obtained at 1 μm pixel size using DMACA sublimation are displayed in Fig. 3a and mass spectra from 5 × 5-pixel regions demarked by the blue and purple crosses are displayed in Fig. 3d. Within the mass spectra a total of 191 lipids were detected in the cerebellar white matter (Fig. 3a; purple cross) and 133 in the cerebellar molecular layer (Fig. 3a; blue cross), as displayed in the pie charts (Fig. 3e, f). Lipid sum compositions were putatively assigned based on a <6 ppm difference from theoretical *m/z* and relative abundances for each class can be found in Supplementary Tables 3, 4. Additionally, CV pre-staining affords the ability to image deoxyribo- or ribo-nucleotides as either intact monophosphate ions or as fragments covalently bound to CV. An overlay of 5 such ions is

displayed in Fig. 3b, and structures and MS² characterisation can be found in Supplementary Figs. 7, 8. Interestingly, the red (adenine) and yellow (uracil) colour channels in Fig. 3b appear highly localised in the Purkinje neuron cell bodies (Fig. 3c; yellow arrows), while purple and cyan colour channels, respectively correspond to thymine and adenine from deoxyribonucleotide origins, are localised to the granule neurons. This highlights the versatility of the source in imaging other analyte classes apart from lipids and reveals how high spatial resolution imaging at a pixel size of 1 μm can improve our fundamental understanding of cell biology by localising specific metabolites to individual cells, in situ.

While sub-micron MSI is notoriously challenging due to the severely reduced sampling volume and subsequently analyte signal intensity, the improvements to sampling efficiency afforded by pre-

staining in combination with cold-plasma ionisation were sufficient to allow for MSI at pixel sizes of 750, 500, 325 and 250 nm (cf. Supplementary Fig. 10). Once the source laser is focussed, reducing laser energy in turn reduces laser ablation spot size, however too substantial a reduction results in minimal sample ejection. Thus, to access pixel sizes <750 nm (i.e. nominal ablation crater size at lowest effective energy setting) oversampling was employed. For example, to achieve 500 nm, the stage was moved by ~30% less than the beam diameter, which only ablates 500 nm of untouched sample material per ejection event. Using this approach to acquire a 500 nm pixel size MS image of the Declive (IV) lobule of mouse cerebellum, a data directed peak list of 53 spatially informative lipids and nucleotides was used to compute a 5-component probabilistic latent semantic analysis (pLSA). Figure 3h displays the spatial distribution of the first three components, with Fig. 3g displaying the corresponding loading spectra. Clear lipid profile differences can be observed between these three components, especially for the third component (blue), which appears to highly correlate to the Purkinje neurons. By using the SCiLS plugin to QuPath, discrete regions corresponding to different cell and tissue types can be precisely generated and exported for mass spectral interrogation. This is displayed in Fig. 3i where enclosed regions were characterised according to El-Azab et al.[36] and correspond to glial cells (yellow) in the outer molecular layer, pyriform Purkinje neurons (green), Bergmann protoplasmic astrocytes (red), granule neurons (blue) and cerebellar white matter (orange). Differences in the averaged mass spectra from each region can be found in the supporting information (cf. Supplementary Fig. 11). Submicron pixel size tissue experiments were additionally conducted on in situ lumbar motor neurons in murine spinal cords at 750 nm and 500 nm (Supplementary Fig. 12) and similarly highlight specific lipid and nucleotide profiles of the motor neurons.

### t-MALDI-P for single- and sub-cellular analyses

A long-standing goal of molecular biology and medical research is to understand how individual cells and cellular organelles respond to certain experimental conditions and stimuli. While MSI provides a solution to this by recording analyte distribution(s) across a coordinate space, single cell—and especially subcellular—MSI is a considerable challenge. Human cells typically range between 4 and 20 μm in size and commercially available MALDI-MSI instruments are currently capable of 5 μm pixel size. To ascertain inter- and intra-cellular molecular information, pixel sizes ≤2 μm are required. This is exemplified in Fig. 4a, which shows the MSI of the human neuroblastoma cell line, SH-SY5Y, at different spatial resolutions. Cells were live-stained with MitoTracker Deep Red and subsequently stained with the nuclear stain Hoechst prior to microscopy (cf. Supplementary Fig. 13 for co-registration investigation). CV staining of the cells was conducted prior to MSI, which both improves contrasting for on-line sample visualisation and MS signal intensity (as per tissues; cf. Fig. 2). Most notable in this image series is the additional cellular details that are gained by improving pixel sizes from 5 μm to 2 μm. At 5 μm, cell nuclei, which are indicated by the purine nucleotides feature at $m/z$ 342 in blue, are spatially convoluted by other lipid signals in the same pixel(s) and thus cannot be distinguished from the surrounding cell. Conversely, at 2 μm, cell nuclei can be clearly distinguished from lipids that share the same spatial organisation as the mitochondrial networks (i.e. 'PC' and PE lipids in red and yellow, respectively). Beyond 2 μm, finer cellular details become more defined and subcellular structures begin to emerge as is best observed in Fig. 4a–c.

Figure 4b displays the 1 μm pixel size MSI and 40× fluorescence microscopy of the osteosarcoma cell line, U2OS, which have undergone DAPI and Nile red staining for microscopy, and CV staining for MSI. Upon irradiation, Nile red emits photons at different wavelengths depending on the polarity of the chemical environment it is within. While in the presence of polar lipids, such as glycerophospholipids,

Nile red emits orange-red wavelength photons, while in the presence of apolar lipids, such as triacylglycerols, it will emit photons in green-yellow wavelengths. Within the 1 μm MS image, four ions were selected to best match the distribution of the fluorescence microscopy channels, including 'PC' 34:2, 'PC' 34:1, 'PC' 36:2 for the lipid rich regions and the CV purine derivative for the nucleus. A magnification of four densely packed cells highlights that these ions are present within localised cell compartments, which are approximately correlated to the different fluorescence channels in the 40× microscopy. Similarly, subcellular spatial correlation can also be observed in Fig. 4c, which shows the aforementioned SH-SY5Y cells imaged through 1.5 μm MSI and 40× fluorescence microscopy. Here, the magnification highlights that individual mitochondria and mitochondrial networks that are observable via the use of MitoTracker Deep red, and specific mass spectral signals (e.g. 'PC' 34:2, PE 34:0, and PE 36:2) are observed in the same regions.

To exemplify the capabilities of the t-MALDI-P source for identifying intercellular molecular differences, we obtained three previously established patient-derived cancer cell lines[37], including: a circulating tumour cell line (UWG02CTC) derived from the peripheral blood and ascites-derived (UWG02ASC) cell line established from peritoneal fluid, taken as simultaneous biopsies from a metastatic gastric adenocarcinoma patient, and a circulating tumour cell line (UWG01CTC) that was derived from a patient with neuroendocrine oesophageal carcinoma. To trace and identify these cells, during monoculture they were labelled with different CellBrite fluorophores. Cells were then co-cultured for 72 h before imaging using 10× fluorescence microscopy and MSI at 750 nm pixel size (Fig. 4d). Within the 750 nm MSI, three lipids were chosen that most differentiated patient line cells. Selecting three representative cells for spectral comparisons, the UWG01CTC cell (blue) appeared to have a higher abundance of longer chain 'PC' 36:1, while both UWG02ASC (red) and UWG02CTC (green) cells originating from the same patient had a higher abundance of shorter (i.e. 'PC' 32:1) and more unsaturated (i.e. 'PC' 34:2) lipids and single-cells were observed to differ in PE 36:1 and PE 36:2 abundance (Supplementary Fig. 14). Nonetheless, assigning MSI generated molecular profiles to specific cells in high-confluency experiments requires MSI spatial resolutions greater than the spacing between the cells and thus the capabilities of the t-MALDI-P source are well suited to high density cell co-culture experiments.

## Discussion

Herein, we highlight the development, capabilities and applications of an AP t-MALDI source that utilises inline plasma ionisation to improve detection sensitivity and analyte coverage capable of achieving submicron pixels sizes for MSI of tissues and cells. The combination of plasma ionisation and sample pre-staining was shown to lead to increased sampling efficiency and ion yields, allowing us to detect up to 200 unique lipid species and nucleotides within 1 μm pixel size MSI experiments. This advance represents significantly more lipid species than previously detected at equivalent resolutions, and additionally detects alternative molecular classes such as nucleotides.

Considering the spatial resolution capabilities, co-registration with fluorescence microscopy—especially involving organelle labelling—has great future potential for molecular cell biology research. By combining these multimodal techniques in parallel with single cells with altered metabolic behaviours (e.g. genetic knockdown, protein inhibition, isotopically labelled tracer experiments, etc.) and image segmentation will inevitably lead to expanded knowledge about the complex interactions between organelles and cellular microenvironments.

The higher spatial resolutions afforded by sample pre-staining in combination with AP-t-MALDI-P allow for increased material ejection, which in turn allows for improved analyte sensitivity at subcellular resolutions. However, assignment of MS spectral profiles to specific

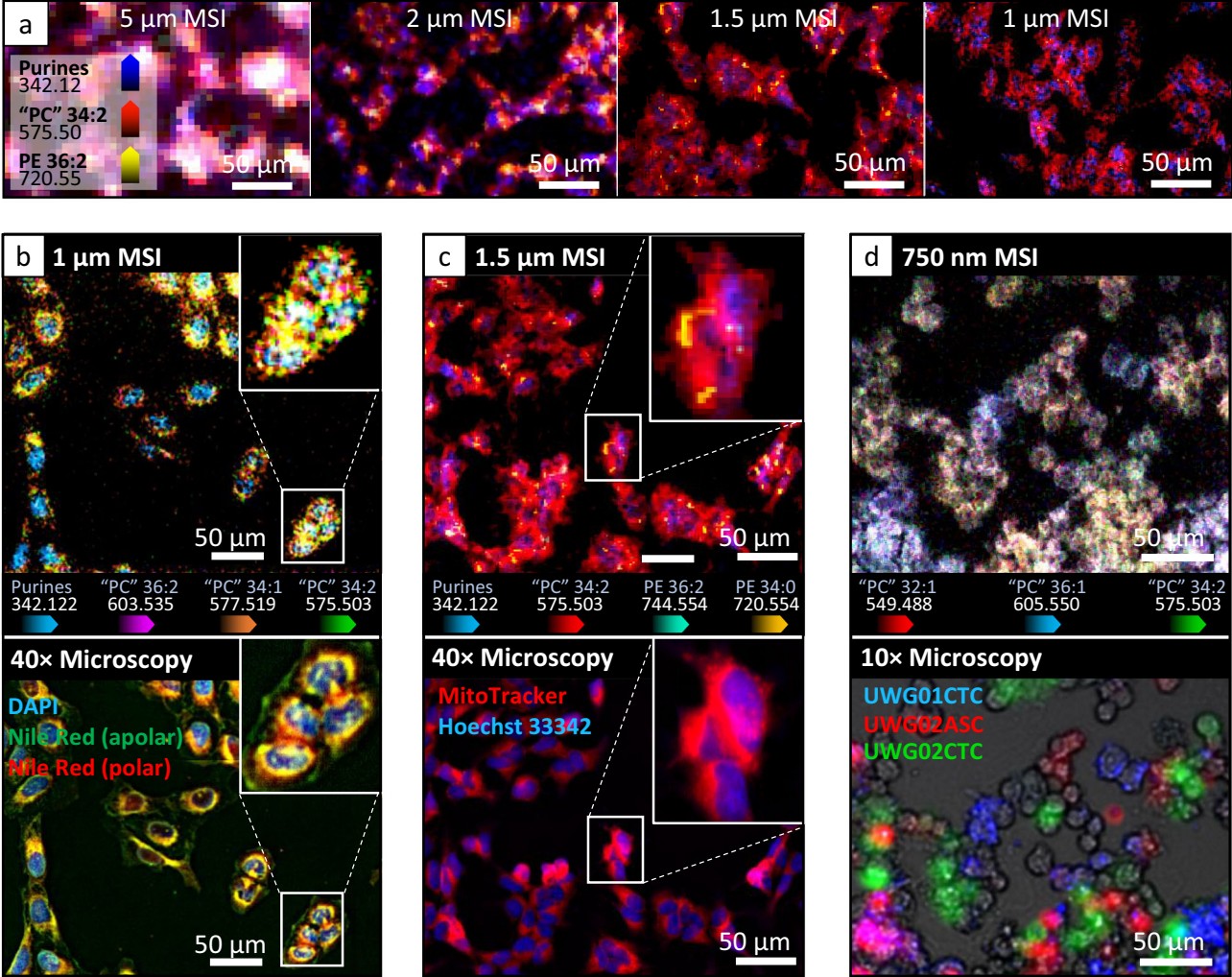

**Fig. 4 | Mass spectrometry imaging of single cell cultures at sub-cellular spatial resolutions. a** Neuroblastoma cell line (SH-SY5Y) imaged through t-MALDI-P MSI at different pixel sizes. MSI pixel sizes of (left-to-right) 5 μm, 2 μm, 1.5 μm and 1 μm display the overlaid ion-distribution images annotated within the first panel. **b** Osteosarcoma cell line (U2OS) stained with DAPI nuclear stain and the lipid stain, Nile red, and imaged through MSI at 1 μm (top) and 3-channel (390/470 nm, 555/550 nm, 475/510 nm) 40× fluorescence microscopy (bottom). **c** Neuroblastoma cell line (SH-SY5Y) stained with the nuclear stain Hoechst and mitochondrial stain

MitoTracker Deep Red and imaged through MSI at 1.5 μm (top) and 2-channel (390/470 nm, 635/700 nm) 40× fluorescence microscopy (bottom). **d** Established patient derived cancer cell lines, where monocultures of each line were stained with different fluorescent CellBrite membrane dyes (red, green, blue) before co-culture. (Top) MSI at 750 nm pixel size, and (bottom) 3-channel (390/470 nm, 555/550 nm, 475/510 nm) 10× fluorescence microscopy. Mass spectral investigations for each single cell MSI can be found in Supplementary Fig. 14. All microscopy images were obtained in triplicate (*n* = 3) and display similar results.

subcellular compartments will require high fidelity co-registration between microscopy and MS images. As such, the onboard fluorescence microscopy co-registration methods described by Potthoff et al.[38], in the same issue of this journal, would significantly improve multimodal imaging co-registration and thus provide the necessary validations for assigning nucleotide and lipid profiles to compartments within the cell.

While the tissue data here were acquired using a standard section thickness of 10 μm (i.e. the approximate diameter of most cells), we note that at these scales minor sample preparation artefacts (such as tissue cracking) can become more pronounced and may warrant new preparation protocols to fully realise the potential of MSI at low-to-submicron scales. Future work may benefit from the use of thinner sections which reduce the probability of sampling multiple tissue or cellular features within a given volume.

Last, while we speculate that ionisation arises by charge transfer of ionised water from the atmosphere to the analyte, the addition of dopant vapours into the plasma source could be explored to access

alternate ionisation processes and selectively enhance different analytes. In summary this work significantly expands the possibilities of MSI for its integration with complementary cellular imaging techniques and addressing many research questions in the field of spatial biology.

## Methods

### Nomenclature

Lipid nomenclature follows previously defined shorthand naming that only states the level of structural detail that has been determined by the conducted analytical techniques[39–41]. In this naming system the lipid category is defined by a two, three or six letter abbreviation (e.g. phosphatidylethanolamine; PE, ceramide; Cer, hexosylceramide; HexCer, etc.) followed by the number of carbons comprising the fatty acyl chain(s) and/or sphingoid backbone and number of carbon-carbon double bonds, separated by a colon (e.g. PE 34:1—a phosphatidylethanolamine containing 34 carbons and one double bond within its fatty acyl chains). Additionally, for sphingolipids the number of

hydroxyl groups attached to the sphingoid backbone is indicated after this, separated by a semicolon (e.g. Cer 36:2;1—a ceramide containing 36 carbons, 2 double bonds and 1 hydroxyl group found within the fatty amide and sphingoid backbone).

Due to known source fragmentation of specific lipid classes, the mass spectral feature is assigned as the majority contributor, but the lipid class is indicated within inverted commas (e.g. 'PC' 38:4—a mass spectral feature assigned as a majority PC 38:4 but spectral intensity could be convoluted by other lipid species known to fragment, such as TGs [major] and other phospholipids [minor]). Previous analysis of lipid standards[32] has revealed the extent of source fragmentation that occurs for each lipid class, and indicates choline containing phospholipids (i.e. phosphatidylcholine; PC, and sphingomyelin; SM) are most affected by source fragmentation and hence are assumed to be majority contributors to this ambiguous spectral feature. Additional structural detail, such as fatty acyl chain composition, relative (*sn*-) positioning around glycerol backbones or regioposition/stereochemistry (*n*-/Δ and *cis/trans*) of double bonds were not determined for this study and thus are not annotated.

## Lipid identification

Developed in-house (cf. Methods: 'Data Processing and statistics'), an automated lipid identification pipeline capable of type I and II mass spectral deisotoping was used to monitor for lipid sum composition species at a user chosen ppm tolerance (<6 ppm) across 14 classes of glycerophospho- and sphingo- lipid were monitored (i.e. PS, PI, phosphatidylethanolamine [PE], phosphatidylglycerol [PG], phosphatidic acid [PA], ether-PE [PE-O], ceramide [Cer], ceramidephosphate [CerP], hexosylceramide [HexCer], dihexosylceramide [Hex2Cer] and trihexosylceramide [Hex3Cer]. As has previously been described with plasma ionisation[32], zwitterionic lipids, such as phosphatidylcholine (PC) and ether-PC (PC-O), undergo in-source fragmentation and loss of the phosphocholine headgroup to a well characterised lipid fragment ion containing a dioxolane ring[2]. Sphingomyelin can also undergo source-fragmentation and loss of the phosphocholine headgroup to form a ceramide (Cer), which is specifically detected as the dehydrated ion, ['Cer'-H$_2$O + H]$^+$ and not as the protonated ion, ['Cer'+H]$^+$. This allows for biologically derived Cer to be detected using the [M + H]$^+$ ion, without possible convolution from SM. Additionally, SM can undergo a separate source fragmentation pathway that leads to the loss of a methyl group from the trimethylammonium, thus this [SM-CH$_3$ + 2H]$^+$ ion can be used to unambiguously assign SM sum composition structure.

These headgroup-loss fragment ions retain fatty acyl sum composition information and hence serve as a proxy for their unfragmented parent lipid. To indicate lipid species identifications that were made using these fragment ions, we have used inverted commas around the lipid headgroup nomenclature to represent this (e.g. 'PC' 34:1). It should be noted that triacylglycerol (TG) lipids, along with other phospholipids can also undergo source fragmentation and loss of a fatty acid to become sum-compositional isomers of the 'PC' fragments. However, with plasma ionisation intact TGs can also be monitored as [M + H]$^+$ cations[32], which were not present within any of the data shown within and thus, to the best of our knowledge, do not contribute here to ion signals characterised as PCs. For further details on lipid assignment criteria, thresholding and logic, cf. Methods: Data processing and statistics.

## Chemicals

The following chemicals were obtained from Sigma Aldrich (Bayswater, Australia) unless otherwise stated. MALDI matrices: 2,5-dihydroxyacetophenone (>97%; D107603), α-cyano-4-hydroxycinnamic acid (>97%; 145505) and 4-(dimethylamino)cinnamic acid (>99%; 218979). Solvents: HPLC grade acetone (>99.9%; 270725), LC-MS grade

water (1.15333), LC-MS grade ethanol (>99.9%; 1.11727) and LC-MS methanol (>99.9%; 1.06035). Other chemicals: cresyl violet acetate (-70%; C5042), Hoechst 33342 (>97%; 14533), ribonucleoside triphosphate set (11277057001), ammonium acetate (>99.99%; 431311) and ammonium fluoride (>98%; 216011). Cell media: Foetal Bovine Serum for U2OS and SH-SY5Y culture was from CellSera Australia (AU-FBS/PG). DMEM/F12 medium (#12500-096), 0.25% Typsin-EDTA dissociation solution (#25200056) and GlutaMAX™ (#35050061) were from ThermoFisher. UWG02CTC, UWG02ASC and UWG01CTC were cultured in Advanced DMEM/F-12 (ThermoFisher) with 10% FCS, 1% penicillin/streptomycin (ThermoFisher), 2 mM L-Glutamine (Sigma) and 0.02 µg/mL EGF (ThermoFisher). Cell culture dyes: CellBrite® green (#30021), red (#30023) and blue (#30024) were obtained from Biotium (California, USA). MitoTracker Deep Red FM (M22425; ThermoFisher), Nile red (19123; >97%; Sigma Aldrich). Consumables: 12-well, removable chamber slides (81201) for cell culture were obtained from ibidi (Gräfelfing, Germany). Glass microscope slides (EM0107) for tissue mounting were obtained from Rowe Scientific.

## Sample preparation

**Tissue sections.** Mouse brains were harvested from healthy mice (NSG NOD.Cg-Prkdcscid Il2rgtm1Wjl; animal sex and age information was not collected) populations produced from other research within the University of Wollongong in accordance with the Australian code for the care and use of animals for scientific purposes. Organs were snap frozen in liquid nitrogen and stored at −80 °C prior to cryo-sectioning. Fresh-frozen brains ($n = 1$) were cryosectioned saggitally to 10 µm thickness ($n = 9$) using a Leica CM1950 cryo-microtome and thaw-mounted on glass microscope slides. Serial sections from approximately the mid sagittal depth were used in this research. Mouse spinal cords (C57BL/6) were embedded in OCT medium at harvest to ensure the cords remained intact during sectioning ($n = 4$), but otherwise follow the same sectioning and mounting protocol as brain sectioning. All animal experiments were approved by the University of Wollongong Animal Ethics Committee (approval number: AEPR22/17) and complied with the Australian National Health and Medical Research Centre code of practice for the care and use of animals for scientific purposes.

**Cell culture.** SH-SY5Y (RRID: CVCL_0019) cells and U2OS (RRID: CVCL_0042) cells were cultured in growth medium (DMEM/F12 media, supplemented with 10% FBS and 1x GlutaMAX™) at 37 °C with 5% atmospheric CO$_2$. Cells were grown to -70% confluency prior to passage or plate out, whereby cells were washed with pre-warmed 1x PBS 1 mM EDTA solution for 2 min prior to a 5 min incubation in 0.25% Trypsin-EDTA dissociation solution. After dissociation, cells were collected and pelleted via centrifugation (300 × *g*, 5 min) prior to resuspension in growth medium. For experiments, cells were plated out in ibibi camber slides ($n = 10$ wells) at -30% confluency and allowed to adhere and proliferate for 48 h prior to staining and fixation. U2OS cells were stained by incubation in serum-free medium containing 500 nM Nile Red for 30 min prior to washing twice with serum-freed medium for 5 min each. SH-SY5Y cells were stained with MitoTracker Deep Red FM by incubation in serum-free medium containing 300 nM Mitotracker dye for 30 min, prior to washing twice in serum free medium for 5 min each. Following staining, cells were fixed using a gentle method to prevent PFA-induced changes to organelle morphology. We added 1 volume of prewarmed 4% (w/v) PFA to the volume of media the cells were incubated in, resulting in a final PFA concentration of 2%. Cells were then left to fix at room temperature for 30 min before they were washed with 0.1 M Tris-buffer (pH 8) for 10 min to quench residual PFA. Nuclear DNA was then stained with a 1:5000 solution of Hoecht 33342 for 10 min with gentle rocking prior to 3 washes in 100 mM aqueous ammonium acetate via dipping before being aired dried overnight for either fluorescence imaging or MSI.

**Patient derived cells.** UWG02CTC, UWG02ASC and UWG01CTC were cultured in Advanced DMEM/F-12 (ThermoFisher) with 10% FCS, 1% penicillin/streptomycin (ThermoFisher), 2 mM L-Glutamine (Sigma) and 0.02 μg/mL EGF (ThermoFisher) and maintained at 37 °C in hypoxic conditions at 3% $O_2$ and 5% $CO_2$. All cells were sub-cultured at ~80% confluence twice weekly, and routinely monitored for the absence of mycoplasma contamination. Short Tandem Repeat (STR) profiling was performed to reconfirm the identity of the cell lines. Cells were enzymatically lifted off the tissue culture flask using TripleE (ThermoFisher). Upon quenching with media cells were pelleted at 300 g for 3 min and resuspended at a density of 300,000 cells/mL. 1 mL of cells were incubated with CellBrite dyes for 20 min at 37 °C in the water bath in 15 mL falcon tubes wrapped in aluminium foil and inverted occasionally to avoid clumping (UWG02CTC−Cellbrite green at 1:400 dilution, UWG02ASC−CellBrite orange at 1:400 and UWG01CTC−CellBrite blue at 1:300). Cells were quenched with 2 mL media, pelleted at 300 g for 3 min and carefully resuspended in 1 mL of media. All three cell types were then mixed in a 1:1:1 ratio and plated into 12-well ibidi chambers at 10,000, 20,000 and 30,000 cells per well densities. Individual stained cell types, as well as unstained cells, were also plated as control for potential wavelength compensation. After 48 h incubation cells were carefully washed twice with PBS (-/-) followed by a brief fixation with 4% PFA for 1 min only. PFA was removed and cells were washed twice in PBS (-/-) followed by three washes in 150 mM aqueous ammonium acetate by dipping. Last, the slide was dried in the tissue culture hood for 5–10 min and then transferred into a slide transport chamber for immediate imaging or vacuum seal and store in −80 °C.

**Histological staining.** Two methods were used for histological staining using cresyl violet. The first is the adapted method, described herein, that retains tissue and cellular lipids during staining. A 1% aqueous solution of cresyl violet acetate was prepared along with a drop (~50 μL) of glacial acetic acid being added for every 50 mL volume. Approximately 10 μL mm$^{-2}$ of solution was pipetted atop the slide mounted tissues and cells, or enough for the sample to be submerged under the droplet bead. After 90 s, the slide was rotated on its side and tapped on a KimTech wipe to remove the stain droplet. Depending on the sample area, ~500 μL of 100 mM aqueous ammonium fluoride was dropwise pipetted across the sample to remove excess stain from the tissue or cells and surrounding areas. The sample was then dried under a gentle flow of nitrogen gas before microscopy, matrix application and MALDI-MSI. This process was tested extensively for any delocalisation of lipids, or deformation and shifting of cells or tissues, however the technique appears to be gentle as no artifacts could be observed (cf. Supplementary Fig. 1). The second method for cresyl violet staining is conventional and was undertaken on samples after MALDI-MSI had been conducted. First, the matrix was washed off the slide by dipping in 100% ethanol for 30 s. The slide was then submerged in 0.2% aqueous cresyl violet solution for 10 min before being submerged in 70% aqueous ethanol for 2 min for differentiation and dehydrated using 95% and 100% aqueous ethanol for 1 min and 3 min, respectively. Xylene clearing was not undertaken as samples were not cover slipped. Brightfield microscopy was then obtained immediately after staining.

**MALDI matrix application.** 5 mg of each matrix (i.e. DHA, CHCA and DMACA) was dissolved in 500 μL of a 1:1 solution of acetone:methanol (v/v) in an Eppendorf tube. The matrix solution was then pipetted onto a steel heat plate within a custom built sublimator device. After the solvents had evaporated, the sample was then placed into the sample holder, which suspends the inverted sample ~5 cm above the matrix heating plate, and is cooled to 10 °C via a Peltier cooler. The vacuum vessel is sealed and pumped down to ~0.1 mBar before the heating plate is activated. Sublimation conditions for each matrix are as follows: DHA, 140 °C for 3 min; CHCA, 180 °C for 5 min; and DMACA, 190 °C for 5 min. The timer was only begun once the set temperature was reached.

**Nucleotide standards experimentation.** 5 μL of 100 mM ribonucleotide triphosphate standards (i.e. adenine, guanine, cytosine and uracil) were mixed in 2 mL glass vials with 10 μL of 1% aqueous cresyl violet acetate solution. Colour changes were noted, and a further 50 μL of 250 mM methanolic dihydroxyacetophenone was added to each vial. 5 × 1 μL aliquots of each solution was then spotted on to a standard glass microscopy slide, allowing the 1 μL droplets to dry before depositing the next. The sample was then run using reflection geometry MALDI-PPI configuration, as previously described by Michael et al.[32], for MS$^1$ and MS$^2$ mass spectrometric analysis.

**AP-t-MALDI-P**
The design was based on the t-MALDI-2 ion source[26], that was instead adapted for AP MALDI and interfacing with a modified timsTOF Pro (Bruker Daltonics) mass spectrometer. A 30 mm optical cage system, including three Ø1" harmonic beam splitters (reflects 355 nm, transmits 532 nm and 1064 nm) and a Ø1" broadband dielectric mirror (reflects 350–400 nm), was constructed to reflect the Nd: YAG 355 nm laser (SmartBeam 3D, Bruker Daltonics) through the rear of a glass slide. In this optical path, the beam diameter was reduced using 8× telescope lensing (Keplerian geometry) and was focussed through a 50× Plan Apo NUV infinity corrected microscopy objective (0.42 NA; Mitutoyo), giving an effective working distance of 15 mm. An aluminium sample plate holder with a Ø40 mm aperture was attached to a stack of three piezoelectric positioners (CLS-92, SmarAct) configured to have 3-axis control (x, y, z) over the sample holder position using an MCS2 controller module (SmarAct), allowing for ±10 nm movement precision. Making use of the reflected visible light through the microscope objective, a CMOS rolling shutter digital camera (IMX290, Sony) was placed behind the dielectric mirror to serve as an online 50× microscope for sample visualisation and laser focus optimisation. On the frontside of the sample plate, a heated (410–420 °C) stainless-steel capillary (OD Ø3.17 mm; ID Ø1.4 mm) was positioned ~1 mm from the samples surface and used the vacuum of the mass spectrometer to draw in the analyte plume to the cold-plasma device (SICRIT, Plasmion) for ionisation and subsequent mass analysis using the timsTOF Pro. Instrument control was performed using a custom developed version of timsControl that allowed for control of the non-standard stage.

**Source parameters**
Once the laser is focussed the ablation crater size is effectively set for a given pulse energy and is highly dependent on the sample type (i.e. cells, tissue, white/grey matter of the brain, etc.). To vary the ablation spot size, the laser energy was adjusted, which alters the fraction of the beam that exceeds the fluence required for material ejection. Laser energy measurements were taken after all laser optics using an energy sensor (QE12HR-H-MB-D0, Gentec-EO), and the values that were used herein to achieve the varied ablation crater sizes are listed below in Table 1. It should be noted that to achieve pixel sizes of less than 750 nm, the optimal laser energy to give the smallest ablation craters (~750 nm) for a given sample type was selected, and the lateral movement of the piezoelectric stage was set to less than the diameter of the beam (i.e. oversampling conditions[42], cf. Supplementary Fig. 15 for further explanation and examples). Mass spectral signal intensity and spectral diversity was significantly reduced at >50% oversampling (e.g. stage movements of 375 nm for a Ø750 nm beam), however lipid and lipid fragment signals were still observed down to 250 nm pixel size (i.e. stage step sizes of 250 nm using oversampling conditions; c.f., Supplementary Fig. 10). The laser was set to pulse 3–20 shots at 10 kHz, with shots being optimised based on sample material type and whether oversampling was deployed.

**Table 1 | Laser energy measurements and resulting ablation spot sizes used for varying sample type**

| Output energy at sample (nJ) | Pulse-to-pulse stability (RMS %) | Mean ablation crater diameter (µm) | Sample type |
|---|---|---|---|
| 90 | 1.84 | 1.0 | SH-SY5Y cells |
| 92 | 2.40 | 1.5 | SH-SY5Y cells |
| 97 | 2.87 | 1.0/0.75 | CTC cells/Tissue |
| 178 | 1.05 | 2.0 | SH-SY5Y cells |
| 203 | 0.88 | 1.0 | Tissue |
| 269 | 0.92 | 2.0 | Tissue |
| 640 | 0.42 | 5.0 | SH-SY5Y cells |

## Mass spectrometry

All mass spectrometric analysis was conducted in positive polarity using a timsTOF Pro mass spectrometer that was modified for AP-t-MALDI-P. A mass range of 250–1500 $m/z$ was selected for MS[1] imaging experiments, with a nominal mass resolving power of ~55,000 at $m/z$ 792 using "focus mode". The ion optics settings for MS[1] experiments were as follows. The deflection 1 delta was set to 100.0 V. The peak-to-peak voltage (Vpp) for the Funnel 1 RF, Funnel 2 RF and Multipole RF was 200 Vpp, 500 Vpp and 850 Vpp, respectively. The quadrupole mass filter was set to an ion energy of 5 eV with a low mass cut off at $m/z$ 250; the collision cell was set to transmit ions using 8 eV and an RF of 2200 Vpp; and transfer time setting of 70 µs with a pre-pulse storage of 12 µs. The TOF voltages were as follows: push 1588.0 V, pull 1588.9 V, fill 61.0 V, extract 642.9 V, lens 6400.0 V, flight tube 9900.0 V, decelerator 639.0 V, reflector 2925.0 V and detector 2189.0 V.

Tandem mass spectrometry experiments (MS[2]) were conducted using a mass range of 50–1500 $m/z$, an isolation width of 1 Da and a collision energy of 15 V. The remaining ion optics and TOF voltages were as described for MS[1], with the following differences. The deflection 1 delta was 70.0 V, and the peak-to-peak voltage for the Funnel 1 RF, Funnel 2 RF and Multipole RF was 250 Vpp, 350 Vpp and 350 Vpp, respectively.

## Microscopy

**Light microscopy.** Brightfield microscopy of tissue sections before and after AP-t-MALDI-P was conducted using a Leica SP8 Falcon confocal microscope. The dry objective lens used was a HC PLAN APO CS2 with 0.40 numeric aperture (NA) for 10× magnification of samples and micrographs were captured using a Leica DFC digital camera using $2 \times 2$ binning and 12-bit digitisation at 40 MHz.

**Fluorescence microscopy.** Brightfield and fluorescence microscopy for cell culture analyses was conducted on a Leica Thunder DMI8 Imaging system. For 40× magnification the dry objective lens used was a HC PLAN APO with a 0.95 NA and the 10× magnification objective was a HC PLAN FLUOTAR with a 0.32 NA. Micrographs were recorded using a Leica DFC9000GTC-VSC11064 digital camera with $2 \times 2$ binning at 16-bit digitisation at 540 MHz. Four LED fluorescence channels (ex./em.: 390/470 nm, 475/510 nm, 555/550 nm and 635/700 nm) were selected based on the cellular fluorescence stain, and channel exposure times (5–250 ms) were varied between non-comparable samples but were kept the same for like-samples. Computational clearing was used for background fluorescence minimisation.

A reporting table with all brightfield and fluorescence microscopy parameters can be found in the Supplementary Data files (cf. Supplementary Data 1—Microscopy Reporting Table).

**Scanning electron microscopy including focussed ion beam milling.** Tissue sections mounted on glass microscopy slides were first coated with a ~20 nm layer of platinum using an Edwards Sputter Coater, with the glass sample slide subsequently being grounded to the SEM carrier plate using silver conductive-adhesive. SEM and FIB experiments were conducted using an FEI Helios Nanolab G3 CX dual beam. SEM was obtained at 2500× magnification for surface micrographs and 5000× cross section micrographs using a current of 0.17 nA, a high voltage of 2.00 kV and an operating pressure of ~5 Pa. The high-performance ion conversion and electron detector was used for secondary electron detection. FIB milling was conducted using an ion beam of Ga[3+] and the sample stage was tilted by 52° for access of the electron beam to the cross-sectional surface.

## Software and image processing

Data collection and imaging region selection(s) were conducted using Bruker Daltonics vendor software: timsControl (v6.0.0-SNAPSHOT 27e73e95) and flexImaging (v7.3 Build 9). MSI data were then compiled using the import functions of SCiLS lab (v2025b Pro), and all processed MS images were subsequently generated through SCiLS Lab using a 15 ppm peak window for display of ion distributions. To best capture absolute ion abundance information across different tissue regions no spectral normalisation was used for tissue MSI, although total ion count (TIC) normalised MSI were observed to be highly similar. TIC normalisation was used for single cell images to account for variability in low intensity mass spectral features. All images use hotspot removal above the 99th percentile. For microscopy images, LAS X Office (v1.4.4.26810) was used to visualise and crop micrographs for figure generation. Multimodal image co-registration was done using the two-point co-registration feature in SCiLS or done manually in Microsoft PowerPoint.

## Data processing and statistics

**Data processing.** Mass spectra, from either entire MS imaged regions or from selected regions of interest (as indicated) were exported from SCiLS Lab with subsequent data processing then being undertaken in Microsoft Excel using VBA scripts that were developed in-house. For sum composition lipid identification (i.e. total carbon units and carbon-carbon double bonds within the combined fatty acids), a VBA 'calculator' was created that uses user-input chemical formulae to generate a lipid library alongside their exact-masses (~1800 species across 12 lipid classes). The logic for inclusion as a putative lipid ID involves four layers of shortlisting, and calculations for ID intensity include two layers of isotopic correction.

- First, the theoretical monoisotopic $m/z$ values are compared against mass spectral data, and lipids are shortlisted if they meet user-input $m/z$ thresholding (< 6 ppm). It should be noted here that only the peak-maxa (i.e. $\frac{dy}{dx} = 0$) are used for comparison to theoretical $m/z$ (i.e. any shoulder features or $0 < \frac{dy}{dx} > 0$ values do not meet inclusion criteria).
- Second, using binomial theorem, the isotope distribution patterns of each shortlisted ID are calculated using the chemical formulae in the library and the theoretical natural abundance of the $^{13}$C isotope. The resulting list of all IDs and isotopes is reordered in ascending order to allow for type-1 isotope correction (i.e. reduction of M + 1 and M + 2 isotope intensities from perceived monoisotopic feature intensities). If the shortlisted lipid ID is calculated to be both (i) isobaric (i.e. <20 ppm) with an

isotope $m/z$, and (ii) the $m/z$ peak intensity falls within 2% of the calculated isotope intensity of the isobaric feature, the lipid ID is subsequently removed from the shortlist. If the $m/z$ feature is calculated to be higher than the calculated isotope intensity, the $m/z$ intensity is subsequently reduced by this value. The ID shortlist then undergoes type-2 isotope correction (i.e. summing together the intensities of naturally occurring isotopes) to provide complete isotope corrected abundances.

- Third, the ID list is further shortlisted using a set of rules that follow lipid trends in eukaryotic biology. It should be noted that these rules, while commonly observed in biology, are more strict than the biological machinery governing lipid diversity. As such, it is quite possible that this step removes a number of lipid identifications that could indeed be present within the sample, but because of being infrequently reported in literature these were classified as low-confidence IDs and were removed from counts. The rules used for shortlisting involve three key principles: (i) even/odd numbered lipid sum compositions, (ii) allowable fatty acyl chain lengths and (iii) number of double bonds allowable for a given fatty acyl chain length. Given that the most common naturally occurring fatty acids are: FAs 12:0, 12:1, 14:0, 14:1, 16:0, 16:1, 18:0, 18:1, 18:2, 18:3, 20:0, 20:1, 20:2, 20:3, 20:4, 20:5 22:0, 22:1, 22:2, 22:3, 22:4, 22:5, 22:6, 24:0 and 24:1, only combinations of these were allowed when assessing the confidence of a given shortlisted ID sum composition. All lipid IDs that could not be theoretically composed of the above fatty acids were removed from the shortlist.

- Last, the resulting list of identifications was used to generate feature lists that could be imported into SCiLS Lab where ion images from each feature were manually checked. Lipid IDs (i.e. mass spectral features) were manually assessed for whether they provided meaningful spatial distributions and lipids that gave low signal-to-noise ion-image (i.e. spatially chaotic ion images that did not appear to correlate to tissue or cell features) were removed from the identification counts.

Using the above automated logic, mass spectral data was processed for lipid identification and semi-quantification (i.e. no internal standards were used) and results were used for the pie graphs and the reported ID counts used throughout the manuscript. While very strict inclusion criteria were used to generate these lipid ID lists, the authors acknowledge that identification based solely on accurate mass MS[1] data is perilous and can result in incorrect assignments. While every effort was made to report only correct identifications, it is possible that some identifications may be incorrect and thus ID counts and quantifications may differ from true values. Nonetheless, even given an incorrect lipid assignment, it should be noted that all assigned $m/z$ features gave ordered MSI, and thus still represent a biological analyte present within the tissue.

**Background subtracted mass spectra.** Where indicated (cf. Supplementary Fig. 11 and Supplementary Fig. 14), background subtraction was used to help with visualisation of the mass spectra. This was conducted by first identifying the lipid and nucleotide features within the mass spectrum (vide supra), conserving this list of $m/z$ values and their respective intensities within the original (x, y) data, and filtering all remaining $m/z$ values before plotting the mass spectrum. Mass spectra were then normalised post-filtering, and as such are labelled with an arbitrary ordinate-axis intensity as they do not represent true-TIC normalised intensities. The vast majority of peaks filtered from the mass spectrum are from PDMS, which occurs at 74 Da repeat intervals across the mass range. While repeat units are far from being isobaric to lipid mass defects (i.e. -500 ppm), their presence can obscure mass spectral visualisation, and thus were removed.

**Statistics.** pLSA was used to characterise the differences in the total lipid profile between regions of tissues or cells. Lipid feature lists, generated from the VBA pipeline (vide supra), were used to inform 5 component pLSA calculations, using the in-built statistical feature in SCiLS Lab. Random initialisation was used for the analysis, with no denoising or scaling factors being used.

## Ethics
All animal experiments were approved by the University of Wollongong Animal Ethics Committee (approval number: AEPR22/17) and complied with the Australian National Health and Medical Research Centre code of practice for the care and use of animals for scientific purposes.

## Reporting summary
Further information on research design is available in the Nature Portfolio Reporting Summary linked to this article.

## Data availability
The data generated in this study have been deposited in Zenodo at https://doi.org/10.5281/zenodo.15477922 or can be obtained from the corresponding author on request. The spectral and graphical data used for figures are provided in the Source Data file. Source data are provided with this paper.

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

## Acknowledgements

S.R.E. acknowledges financial support the Australian Research Council (FT190100082) and the Human Frontiers Science Program Organization (RGP0002/2022). L.M. acknowledges support from MNDA, FightMND, and NHRMC. JSL acknowledges financial support from Motor Neuron Disease Research Australia in the form of a Bill Gole Postdoctoral Fellowship (PDF2307). The authors wish to thank Jens Hoehndorf (Vice President Instrumentation R&D, Life Science Mass Spectrometry, Bruker Daltonics GmbH & Co KG) for his support and technical input throughout and Adam Gallaty for his assistance with manufacture of the heated capillaries. The authors acknowledge use of facilities within the UOW Imaging Facility, and the facilities and assistance of Mitchell Nancarrow at the UOW Electron Microscopy Centre.

## Author contributions

Conceptualisation: M.N., J.S., S.R.E., R.S.E.Y. Methodology: R.S.E.Y., J.S., M.N., S.R.E. Software: R.S.E.Y., M.N. Validation: R.S.E.Y., A.K.P., L.M. Formal Analysis: R.S.E.Y. Investigation: R.S.E.Y., J.C.M., A.K.P., L.M., J.S.L. Resources: A.K.P., L.M., J.S.L., S.R.E. Data Curation: R.S.E.Y. Writing (original draft): R.S.E.Y. Writing (editing): S.R.E., J.S., M.N., A.K.P., J.C.M., J.S.L., L.M. Visualisation: R.S.E.Y. Supervision: S.R.E., M.N., J.S. Project administration: S.R.E., M.N. Funding acquisition: S.R.E.

## Competing interests

M.N. is a research and development physicist for Bruker Daltonics GmbH & Co KG, Bremen, Germany who manufactured the timsTOF mass spectrometer used in this study. All remaining authors report no competing interests.
