## [Transparent Peer Review file · Nature Communications]

Subcellular mass spectrometry imaging of lipids and nucleotides using transmission geometry ambient laser desorption and plasma ionisation

Corresponding Author: Dr Shane R. Ellis

Version 0:

Reviewer comments:

Reviewer #1

(Remarks to the Author)

The work is exciting and demonstrates enhanced spatial resolution and several interesting enhancements to sensitivity. The work shows excellent MSI examples at unprecedented spatial resolution. Overall, this is beautiful work.

The following points, if addressed, could strengthen the presentation:

1) More discussion on why the prestaining changes the tissue properties would be nice. For example, is it caused by the stain itself or the interaction of the tissue with the solvents (submerging the sample under a drop of 1% CV acetate solution and removing the excess with aqueous ammonium fluoride)? Is this similar to other approaches of "fixing" samples with various solvents before imaging them (and this then changes the MALDI characteristics)? As this is a key development of the new approach, understanding what causes the enhancement would be useful.

2) The comment is made that reducing laser power reduces the spot size. While the 1/e spot size is reduced as power is reduced (and one could observe this using the ablation crater formed), it is not clear this really reduces the irradiated area as small as the claimed 250 nm spot and perhaps could be partially an artifact of a power threshold needed for observing MALDI crater. Several have reported MALDI spectra from tissues at laser energies below the threshold required for tissue damage / ablation craters. This point is important as the ablation crater dimensions are what the authors use to report the spot sizes / amazing resolutions reported. (Think of confocal and other optical microscopies: people don't reduce laser power to increase spatial resolution many fold but change the optics to focus the laser).

3) The complete ablation approach appears effective in this application to reduce effective new area sampled by their laser. However, this does not produce spot sizes similar to a reduced laser spot diameter but an elongated crescent shaped spot that is considerably larger than a circular spot of the same reported dimension. This point was not discussed (perhaps cite prior papers that discuss this)?

The high resolution MSI images of the tissues are amazing. It appears that all tissue sections were 10 microns thick even for the smallest pixel sizes. As the XY dimensions are reduced, one expects they would have to reduce the tissue thickness. It is not clear how sampled depth changes as spot size is reduced. While they partially addressed with the depth profiling studies, this point is still not well addressed. How does optimal sample preparation requirements / thickness change as the resolution is reduced? When do sample preparation artifacts become limiting? Can the authors estimate the actual sampled volume (given the combination of 10 micron thick tissues, complete ablation with crescent shaped spots and focused laser, it is hard to determine the estimated volume sampled or the minimum amount of material required to have a detectable signal).

Reviewer #2

(Remarks to the Author)

The manuscript entitled "Subcellular mass spectrometry imaging of lipids and nucleotides using transmission geometry ambient laser desorption and plasma ionisation" by Reuben S.E. Young is an exciting work, very timely coinciding with several works advancing MALDI-MSI in lipids, and it supposes a significant qualitative step in

increasing the image resolution typically of 5-10 μ m to 1 μ m and even with valuable data at 250nm, entering to subcellular level. The manuscript also highlights other points, such as utilizing plasma ionization to incorporate more lipid species, and the special case of imaging nucleotides, as these react differently to the staining method, resulting in a greater MS signal.

The manuscript is very exciting. However, some of the claims are somewhat confusing, and the reviewer believes the manuscript would benefit from clarifying these points.

Concerns

1. Figure 2 compares the 'conventional' method with the 'pre-staining' method. However, in this comparison, it is unclear whether figures 2d, 2e, f, and 2h were obtained under the same experimental conditions. The figure legend for 2e states, "MSI at 2 μ m pixel size." In the main text, 2e is also mentioned to be at 2 μ m MSI. The authors were then surprised by the actual ablation crater size (Figure 2dv, which showed a size of 0.7 μ m). Consequently, the laser was adjusted to maximize the ablated material from a 2 μ m pixel, resulting in a crater size of \leq 2 μ m. After this adjustment, Figure 2h was generated. So, it sounds that 2dv and 2e,f are the same conditions, but 2h uses the new conditions. This should be explained clearly, stated, or show a 2h signal before and after adjusting the laser energy.

2. Figures 2g and 2h support two findings. First, the 10X MS signal increases with elevated laser energy, and Second, the number of detected species increases. Since more energy is applied, it is possible that the fragmentation of lipids has also increased, and the newly found species are fragments of more complex lipids (for example, SM giving ceramides, or phospholipids generating DAGs). That would be a disadvantage more than an advantage. Some data and discussion on this point would be helpful.

3. The colors in Figures 2g and h do not help identify the lipids in the pie graph. Some colors are very similar.

4. Supplemental Figure 2 explains that cresyl violet helps the laser to excite not only the interphase with the matrix but also the molecules in the path of the laser beam. This effect differs from the findings in Figure 2, where CV enables smaller craters and allows the laser energy to be increased to achieve equal crater size. Are these two effects of CV complementary? These should be clarified.

5. Supplemental Figure 7: Different matrices are compared. However, the effect of CV is evaluated, or at least is not shown. Each matrix can have different effects on crater size, ionization, and signal intensities for different lipids. The figure compares these matrices using the 'pre-staining' method. However, the effect of CV on improving signal detection in each matrix is not shown.

Version 1:

Reviewer comments:

Reviewer #1

(Remarks to the Author)

As stated before, the work is well performed and exciting. The authors have addressed the minor comments of the reviewers well and the changes do a nice job in addressing them.

The new figure in the SI showing the effective pixel sizes from the oversampling are clear and well done. The reviewer agrees with the comment from the authors that improved sampling will be needed. After all, SEM and other approaches have long noted the differential drying and shrinkage of distinct brain regions containing more cell soma versus axons.

There are no further comments to this exciting work.

Reviewer #2

(Remarks to the Author)

The authors have successfully answered all the concerns. My suggestion is to accept for publication.

Reviewer #1 (Remarks to the Author):

R1.0. The work is exciting and demonstrates enhanced spatial resolution and several interesting enhancements to sensitivity. The work shows excellent MSI examples at unprecedented spatial resolution. Overall, this is beautiful work.

The Authors would like to thank Reviewer #1 for their kind comments and thorough review of the presented work and are happy to hear they share our excitement for this work. Through their comments it is apparent that they have comprehensive knowledge on the subject matter and as such inclusion of the suggested materials will undoubtedly strengthen the manuscripts quality and clarity. Further details about the specific amendments to the manuscript are addressed below under each comment.

The following points, if addressed, could strengthen the presentation:

R1.1. More discussion on why the prestaining changes the tissue properties would be nice. For example, is it caused by the stain itself or the interaction of the tissue with the solvents (submerging the sample under a drop of 1% CV acetate solution and removing the excess with aqueous ammonium fluoride)? Is this similar to other approaches of ‘fixing’ samples with various solvents before imaging them (and this then changes the MALDI characteristics)? As this is a key development of the new approach, understanding what causes the enhancement would be useful.

The Reviewer raises a good point. The signal intensity enhancement observed within the current manuscript is largely accounted for by the increase to sampling volume from desorption of the whole tissue in the beam path, enabled by the pre-staining approach. This is apparent by the additional loss of sample material observed in the FIB-SEM and microscopy images ([previous version] Supplementary Figures 2-3). To demonstrate that enhancement is driven largely by the addition of CV to the tissue, we have now included an additional supplementary figure displaying MSI of murine brain tissue, with and without prestaining, but instead using water to wash the CV away. The below figure is now provided as Supplementary Figure 2 and is referred in the main text as follows:

“To exclude that this effect was related to the ammonium fluoride salt wash, Supplementary Fig. 2 demonstrates that the same signal intensity increase can be observed even when using a water wash for CV stained tissues.” ([new version] Page 5, lines 25-27).

Supplementary Fig. 2: Exclusion of the ammonium fluoride wash leading to the majority of signal enhancement effects during sample pre-staining.

(Left) 2 μm MSI of mouse cerebellum, with no CV prestaining and using DHA as the matrix. (Right) 2 μm MSI of mouse cerebellum, employing the CV prestaining technique, however water was used instead of aqueous ammonium fluoride to wash away excess CV from the tissue. Marked increases in all lipid related MS signals can be observed (10-20 fold), demonstrating that the signal enhancement effects can largely be attributed to the addition of CV to the tissue layer as opposed to salt washing effects. Spectra are of the whole imaging regions, and are non-normalised with the same y-scale for ease of comparison.

The motivation of the ammonium fluoride wash arose due to the well know benefits of tissue washing with ammonium salts to reduce adduct formation and recent work demonstrating the benefit of ammonium fluoride washing for lipid imaging using MALDI (Holbrook *et al*, *Anal. Chem*, 2023, 10603). The systematic exploration of washing and dopants and their effect on lipid detection using plasma ionisation is a topic we are actively exploring further but lies outside the scope of this manuscript.

R1.2 The comment is made that reducing laser power reduces the spot size. While the 1/e spot size is reduced as power is reduced (and one could observe this using the ablation crater formed), it is not clear this really reduces the irradiated area as small as the claimed 250 nm spot and perhaps could be partially an artifact of a power threshold needed for observing MALDI crater. Several have reported MALDI spectra from tissues at laser energies below the threshold required for tissue damage / ablation craters. This point is important as the ablation crater dimensions are what the authors use to report the spot sizes / amazing resolutions reported. (Think of confocal and other optical microscopies: people don't reduce laser power to increase spatial resolution many fold but change the optics to focus the laser).

While the lasers 1/e beam diameter is independent of pulse energy (as it represents the distance needed to reduce by a factor of 1/e), the size of the ablation crater can be controlled by the pulse energy. However, a certain fluence threshold is needed to initiate material ejection in transmission mode (noting that is this a different phenomenon than conventional reflection mode MALDI). Due to the Gaussian profile of the laser, as the pulse energy is increased so too does the fraction of the beam width that exceed the threshold needed for material ejection, and thus analyte detection. Changes to text are outlined below in red (black is pre-existing):

“Once the laser is focussed the ablation crater size is effectively set for a given pulse energy and is highly dependent on the sample type (i.e., cells, tissue, white/grey matter of the brain etc.). To vary the ablation spot size, the laser energy was adjusted, which alters the fraction of the beam that exceeds the fluence required for material ejection.” [new version] Page 17, line 21-23).

The Reviewer is correct in saying that craters of 250 nm cannot be obtained with the current setup. As shown in Figure 2bv, the optimal crater size for signal detection is ~ 750 nm. Reducing the pulse energy further leads to a rapid reduction in signal. As outlined in the manuscript the 500 nm and 250 nm pixel size data was acquired using the oversampling approach whereby all material is depleted from a given spot and then the stage is moved by a distance smaller than the laser diameter. To clarify this, we have attempted to use 'pixel size' throughout to represent what the stage movement distance was, and 'ablation crater size/diameter' to represent the effective sampling volume. The text explaining this is quoted below:

“It should be noted that to achieve pixel sizes of less than 750 nm, the optimal laser energy to give the smallest ablation craters (~ 750 nm) for a given sample type was selected, and the lateral movement of the piezoelectric stage was set to less than the diameter of the beam (i.e., oversampling conditions). Mass spectral signal intensity and spectral diversity was significantly reduced at >50% oversampling (e.g., stage movements of 375 nm for a $\varnothing 750$ nm beam), however lipid and lipid fragment signals were still observed down to 250 nm pixel size (i.e., stage step sizes of 250 nm using oversampling conditions; cf., Supplementary Fig. 10).” ([new version] Page 17, line 25-31).

R1.3.i. The complete ablation approach appears effective in this application to reduce effective new area sampled by their laser. However, this does not produce spot sizes similar to a reduced laser spot diameter but an elongated crescent shaped spot that is considerably larger than a circular spot of the same reported dimension. This point was not discussed (perhaps cite prior papers that discuss this)?

We assume here the Reviewer is referring to the 500 nm and 250 nm data acquired using oversampling (*i.e.*, complete ablation approach). Within oversampling experiments the first row of pixels will be from an ablated area similar to a crescent shape and will present as a higher signal intensity than the rest of the experiment. The subsequent rows of pixels will all be almost a half-crescent shape due to ablated area now being oversampled from both x and y directions. This newly ablated area is equal to a square pixel of the same dimensions as the step movement. To clarify this point a new graphic has been added to the Supplementary Information (Figure S15; *vide infra*) to visually depict oversampling, and an additional early reference from Jurchen *et al.*, 2015 [41] on MALDI oversampling (DOI: 10.1016/j.jasms.2005.06.006) has now been added in red to the text as follows (black is pre-existing):

*“It should be noted that to achieve pixel sizes of less than 750 nm, the optimal laser energy to give the smallest ablation craters (~750 nm) for a given sample type was selected, and the lateral movement of the piezoelectric stage was set to less than the diameter of the beam (*i.e.*, oversampling conditions,⁴¹ cf. Supplementary Fig. 15 for further explanation and examples).”* ([new version] Page 17, lines 25-28).

Supplementary Figure 15: MALDI oversampling spot size, shape, area and sampling volume.

‘Oversampling’ is a term describing the movement of the sample stage a distance less than the diameter of the ablation crater created by the laser. For example, in the above figure (top of graphic) the hypothetical ablation crater is Ø750

nm, however the stage is set to move only 250 nm in the x-direction. If complete ablation is achieved (i.e., additional laser shots give no additional analyte signal) for each stage movement, then analyte signal can only arise from the newly revealed, unablated 'crescent' area without convolution by analyte signal from previously ablated areas. This method can effectively reduce the spatial resolution of MALDI-MSI, without the need for specialised optics to create smaller beam diameters.

As oversampling occurs in both the x- and y-directions during an MSI experiment, this leads to a difference in spot size and shape between the first row and all subsequent rows. After the first row has been completed and the second row begins, the unablated area now resembles a half-crescent shape, as is displayed in the above figure (bottom of graphic). Again, using the example of a $\varnothing 750$ ablation crater with a stage movement of 250 nm, this half crescent ablation spot has the equivalent area to that of a 250 nm square pixel, however the newly ablated area does span twice the distance (500 nm) in the x-direction (assuming equal ablation across the region), but only once the distance (250 nm) in the y-direction. Given a wet-tissue section of standard thickness for MALDI-MSI (10 μm) and a tissue density near that of water, the ablated volume of a 250 nm pixel would be approximately equal to 625 fg of material.

R1.3.ii. The high resolution MSI images of the tissues are amazing. It appears that all tissue sections were 10 microns thick even for the smallest pixel sizes. As the XY dimensions are reduced, one expects they would have to reduce the tissue thickness. It is not clear how sampled depth changes as spotsize is reduced. While they partially addressed with the depth profiling studies, this point is still not well addressed. How does optimal sample preparation requirements / thickness change as the resolution is reduced? When do sample preparation artifacts become limiting? Can the authors estimate the actual sampled volume (given the combination of 10 micron thick tissues, complete ablation with crescent shaped spots and focused laser, it is hard to determine the estimated volume sampled or the minimum amount of material required to have a detectable signal).

This is correct and an astute observation. Indeed, the cylindrical sampling volume now has a height (sample thickness) greater than the diameter (XY step size). Within the FIB-SEM experiments (Supplementary Figure 3 [previous version]) it can be observed that while the tissue was sectioned at 10 μm thickness, desiccation of the sample during normal storage and sample preparations leads to it reducing in thickness to about 1-3 μm . Moreover, due to the staining method employed the entire volume of the tissue is sampled, rather than just the matrix layer. Here we used 10 μm sections as these are approximately equal to the thickness of most cells, which is standard for MSI. The Reviewer is also correct in noting that as resolution is improved so too does the effect of tissue preparation artefacts. We agree and have now added the following paragraph to the manuscript to highlight this, and the importance for future, to the reader:

“While the tissue data here were acquired using a standard section thickness of 10 μm (i.e., the approximate diameter of most cells), we note that at these scales minor sample preparation artefacts (such as tissue cracking) can become more pronounced and may warrant new preparation protocols to fully realise the potential of MSI at low-to-submicron scales. Future work may benefit from the use of thinner sections which reduce the probability of sampling multiple tissue or cellular features within a given volume.” (Page 12, lines 3-7 [new version]).

Regarding the sampled volume, in the instance of a 250 x 250 nm pixel with a wet-sample section thickness of 10 μm , the sample volume (assuming the tissue has the density of water) would be approximately 625 fg. To assist in estimating sampling volume, this has been added to the caption of the newly added Supplementary Figure 15 [new version], displayed in response R1.3.i.

Reviewer #2 (Remarks to the Author):

R2.0. The manuscript entitled “Subcellular mass spectrometry imaging of lipids and nucleotides using transmission geometry ambient laser desorption and plasma ionisation” by Reuben S.E. Young is an exciting work, very timely coinciding with several works advancing MALDI-MSI in lipids, and it supposes a significant qualitative step in increasing the image resolution typically of 5-10 μ m to 1 μ m and even with valuable data at 250nm, entering to subcellular level. The manuscript also highlights other points, such as utilizing plasma ionization to incorporate more lipid species, and the special case of imaging nucleotides, as these react differently to the staining method, resulting in a greater MS signal.

The manuscript is very exciting. However, some of the claims are somewhat confusing, and the reviewer believes the manuscript would benefit from clarifying these points.

The Authors would like to thank Reviewer #2 for their enthusiasm in support of our work. Their thorough review and comments have expertly captured the central ideas of the manuscript and their comments and suggestions for change will incontrovertibly assist in improving the manuscripts quality and clarity. Detailed responses to each comment are found below.

R2.1. Figure 2 compares the ‘conventional’ method with the ‘pre-staining’ method. However, in this comparison, it is unclear whether figures 2d, 2e, f, and 2h were obtained under the same experimental conditions. The figure legend for 2e states, “MSI at 2 μ m pixel size.” In the main text, 2e is also mentioned to be at 2 μ m MSI. The authors were then surprised by the actual ablation crater size (Figure 2dv, which showed a size of 0.7 μ m). Consequently, the laser was adjusted to maximize the ablated material from a 2 μ m pixel, resulting in a crater size of \leq 2 μ m. After this adjustment, Figure 2h was generated. So, it sounds that 2dv and 2e,f are the same conditions, but 2h uses the new conditions. This should be explained clearly, stated, or show a 2h signal before and after adjusting the laser energy.

The Reviewer has correctly captured the experiments at hand. Indeed, the laser parameters used to create the ablation array for the conventional preparation SEM (Fig. 2av) are the same laser parameters used for the subsequent 2 μ m MSI (Fig. 2b). This is in contrast to the SEM displayed in Fig. 2dv, which uses the same laser parameters to create the array on the pre-stained tissue as for the SEM of the conventionally prepared tissue in Fig. 2av. Because of the reduced ablation crater diameter afforded by using pre-staining, we were thus capable of increasing the laser energy to increase the diameter of the ablation crater closer to the step-size (*i.e.*, 2 μ m). While this was explained in text, the Authors agree that text could be altered to assist with clarity of these details. These additions and changes are highlighted below in red.

“Concurrently, to investigate ablation crater size for the conventional preparation approach, 5 μ m-spaced ablation arrays were obtained using MSI relevant laser energies (cf. Methods section) for scanning electron microscopy (SEM), which revealed \sim 1.75 μ m wide ablation craters (Fig. 2a v).” (Page 3, line 18 – Page 4, line 3 [new version]).

AND:

*“Surprisingly, when using identical laser energies as the conventional preparation to generate 5 μ m-spaced ablation arrays for the pre-staining approach, SEM revealed notably smaller ablation crater sizes from the pre-stained sample (Fig. 2d v; \sim 750 nm). Thus, the laser energy was adjusted to maximise the ablated material from a 2 μ m pixel (*i.e.*, ablation crater size \leq 2 μ m). Increasing laser energy, while maintaining comparable ablation crater diameters, yielded a pronounced enhancement in analyte ion yield – up to an order of magnitude – evident in the increased mass spectral signal intensity observed in the pre-stained tissue sample (Fig. 2h) relative to the conventionally prepared sample (Fig. 2g).” (Page 5, lines 19-25 [new version]).*

R2.2. Figures 2g and 2h support two findings. First, the 10X MS signal increases with elevated laser energy, and Second, the number of detected species increases. Since more energy is applied, it is possible that the fragmentation of lipids has also increased, and the newly found species are fragments of more complex lipids (for example, SM giving ceramides, or phospholipids generating DAGs). That would be a disadvantage more than an advantage. Some data and discussion on this point would be helpful.

While the Reviewer is indeed correct that increased laser power can (and does) lead to increased fragmentation and potential mis-identification of lipids, there were multiple precautions set in place to ensure mis-identification was not possible. The multi-layer identification logic is discussed in detail within the methods section (Data processing and statistics; *cf.* Page 19, line 11). Regarding the example used by the Reviewer – sphingomyelin (SM) fragmentation leading to identical structures to those of ceramides (Cer) – we were careful in selecting ions that do not arise from fragmentation. For example, SM does fragment to Cer, however in this system it seemingly never fragments to give the $[M+H]^+$ feature, only giving rise to ions of equivalent m/z as ceramides $[M-H_2O+H]^+$ features. Thus, Cer identifications were made solely using $[M+H]^+$ m/z values. SM also undergoes a separate demethylation fragmentation, and thus SM species were assigned using m/z values of $[M-CH_3+2H]^+$. An additional point to note is that the pie charts, while presenting that pre-staining does give rise to more identifications, display that the relative proportions of each lipid class wedge are roughly still equal, with the exception of HexCer. This would indicate that we are improving detection of all lipid classes approximately equally, instead of one class fragmenting into another and being misassigned as the wrong lipid.

While previously explored within the stated citation (<https://doi.org/10.1021/acs.analchem.2c03745>), the Authors agree that this information is highly relevant to assuring the reader of the validity of the data. Source fragmentation was discussed in brief within the methods section, as this has been examined in previous publications, however this has now been amended to include more detail, including the above explanations and examples. The following additions in red have been made to text (black is pre-existing):

“As has previously been described with plasma ionisation,[32] zwitterionic lipids, such as phosphatidylcholine (PC), and ether-PC (PC-O), undergo in-source fragmentation and loss of the phosphocholine headgroup to a well characterised lipid fragment ion containing a dioxolane ring.[2] Sphingomyelin (SM) can also undergo source-fragmentation and loss of the phosphocholine headgroup to form a ceramide (Cer), which is specifically detected as the dehydrated ion, $[“Cer”-H_2O+H]^+$, and not as the protonated ion, $[“Cer”+H]^+$. This allows for biologically derived Cer to be detected using the $[M+H]^+$ ion, without possible convolution from SM. Additionally, SM can undergo a separate source fragmentation pathway that leads to the loss of a methyl group from the trimethylammonium, thus this $[SM-CH_3+2H]^+$ ion can be used to unambiguously assign SM sum composition structure.” (Page 13 Line 29 – Page 14, line 2 [new version]).

R2.3. The colors in Figures 2g and h do not help identify the lipids in the pie graph. Some colors are very similar.

The Authors agree that some colours are difficult to differentiate. With this in mind, the chosen colours in the Figure 2 Pie charts (as well as Fig. 3 pie charts) have now been amended.

R2.4. Supplemental Figure 2 explains that cresyl violet helps the laser to excite not only the interphase with the matrix but also the molecules in the path of the laser beam. This effect differs from the findings in Figure 2, where CV enables smaller craters and allows the laser energy to be increased to achieve equal crater size. Are these two effects of CV complementary? These should be clarified.

What we speculate is happening is that the CV is absorbing some energy from the laser beam, and thus only the high energy centre of the gaussian is capable of penetrating through to the tissues' surface. The lower energy 'tails' of the gaussian distribution curve are instead absorbed by the CV, and are released as heat energy to the surrounding tissue. Thus, there is a thermal desorption process that is occurring additional to the MALDI process. Increasing the laser power just widens the range around the centre of the gaussian that is capable of penetrating the tissue, and thus increases ablation crater size. However, at this moment this is just a hypothesis and we are still trying to find the correct experiments to support this for a more in-depth study on how the UV laser is interacting with CV and other stains. This has now been added to the figure caption of Supplementary Fig. 3 (previously Supplementary Fig. 2) and changes to text are displayed below in red (black is pre-existing):

“When the tissue does not contain CV, the laser light transmits through the non-photoactive tissue into the matrix layer, leading to ejection of matrix-lipid co-crystals and the formation of a MALDI plume (G-left). Instead, when CV is present within the tissue, we hypothesise that the CV molecules in the path of the laser beam absorb some of the laser energy. Thus, only the high energy centre of the gaussian is capable of penetrating through to the tissues' surface for material ejection. The lower energy 'tails' of the gaussian distribution curve are instead absorbed by the CV, and are released as heat energy to the surrounding tissue (F-right). Thus, there is a thermal desorption process that causes the ejection of a 'column' of material in addition to the MALDI process (G-right).” (Supplementary Information Page 6, lines 2-6 [new version]).

R2.5. Supplemental Figure 7: Different matrices are compared. However, the effect of CV is evaluated, or at least is not shown. Each matrix can have different effects on crater size, ionization, and signal intensities for different lipids. The figure compares these matrices using the 'pre-staining' method. However, the effect of CV on improving signal detection in each matrix is not shown.

We agree with the Reviewer that each matrix can have different effects on ablation crater size and ionisation efficiencies, as is partially observed in the FIB-SEM data displayed in Supplementary Figure S3F-K [previous version]. As the Reviewer notes, we do display the overall signal intensity differences between each of the three matrices in combination with CV pre-staining (Supplementary Figure 7 [previous version]), however comparison between conventional and pre-staining methods for each matrix was absent. The effect of CV on individual matrices appeared to be quite similar (*i.e.*, increasing MS signal by approximately an order of magnitude) and thus was originally omitted. To demonstrate the CV pre-staining effect on MS signal intensity is independent from matrix choice, we have now included a new Supplementary Figure (Supplementary Fig. 5 [new version]), which is also displayed overleaf.

This Figure is now referenced in-text as follows:

“The effect of improved sampling efficiency becomes apparent when contrasting the averaged mass spectra, which reveal up to a 10-fold increase in ion intensity from the pre-stained tissue (Fig. 2h) compared to the unstained tissue (Fig. 2g) – an effect that was reproducible for alternative matrix choices (cf. Supplementary Fig. 5).” (Page 6, lines 1-4 [new version]).

Supplementary Fig. 5: Comparison of mass spectral intensities between conventional and pre-staining preparations using alternative matrix choices (α -CHCA and 2,5-DHA).

Panels A-D display 2 adjacent 10 μ m thick section of mouse hippocampus, imaged using 10 \times brightfield microscopy and 2 μ m pixel MSI (5 mg α -CHCA matrix), and associated mass spectra from the whole imaged regions. (A) MSI and microscopy were obtained using the 'conventional' preparation method (*cf.* Main Text Fig. 2a), and the MSI displays the overlaid ion-images of 6 lipids indicated in the key below. (B) Absolute intensity mass spectrum averaging the mass spectra from each pixel of the imaged region in (A). (C) MSI and microscopy were obtained using the "pre-staining" preparation method (*cf.* Main Text Fig. 2d), and the MSI displays the overlaid ion-images of 6 lipids indicated in the key below. (D) Absolute intensity mass spectrum averaging the mass spectra from each pixel of the imaged region in (C). As is observed by the comparison of (B) and (D), the addition of CV from the pre-staining method leads to 5-10 fold increase in MS signal intensity when using α -CHCA as the matrix.

Panels E-H display 2 adjacent 10 μ m thick section of mouse hippocampus, imaged using 10 \times brightfield microscopy and 2 μ m pixel MSI (5 mg 2,5-DHA matrix), and associated mass spectra from the whole imaged regions. (E) MSI and microscopy were obtained using the 'conventional' preparation method (*cf.* Main Text Fig. 2a), and the MSI displays the overlaid ion-images of 6 lipids indicated in the key below. (F) Absolute intensity mass spectrum averaging the mass

spectra from each pixel of the imaged region in (E). (G) MSI and microscopy were obtained using the “pre-staining” preparation method (*cf.* Main Text Fig. 2d), and the MSI displays the overlaid ion-images of 6 lipids indicated in the key below. (H) Absolute intensity mass spectrum averaging the mass spectra from each pixel of the imaged region in (G). As is observed by the comparison of (F) and (H), the addition of CV from the pre-staining method leads to 10-20 fold increase in MS signal intensity when using 2,5-DHA as the matrix.